



# Emission characteristics of reactive organic gases from industrial volatile chemical products (VCPs) in China

**Sihang Wang[1,2], Bin Yuan[1,2,*], Xianjun He[1,2], Ru Cui[1,2,a], Xin Song[1,2], Yubin Chen[1,2], Caihong Wu[1,2], Chaomin Wang[1,2], Yibo Huangfu[1,2], Xiaobing Li[1,2], Boguang Wang[1,2], Min Shao[1,2]**

[1] Institute for Environmental and Climate Research, Jinan University, Guangzhou 511443, China

[2] Guangdong-Hongkong-Macau Joint Laboratory of Collaborative Innovation for Environmental Quality, Guangzhou 511443, China

[a] now at: Nanjing Intelligent Environmental Science and Technology Co.Ltd, Nanjing 211800, China

*Email: byuan@jnu.edu.cn



## Abstract:

Volatile chemical products (VCPs) have become an important source of reactive organic gases (ROGs) in urban areas worldwide. Industrial activities can also utilize a large amount of VCPs and emit many organic gases into the atmosphere. Due to multiple sampling and measurement challenges, only a subset of ROG species is usually measured for many industrial VCP sources. This study aimed to investigate the emissions of ROGs from five industrial VCP sources in China, including shoemaking, plastic surface coating, furniture coating, printing, and ship coating industries. More comprehensive speciation of ROG emissions from these industrial VCP sources was developed by the combination of the proton transfer reaction time-of-flight mass spectrometer (PTR-ToF-MS) along with gas chromatography-mass spectrometer/flame ionization detector (GC-MS/FID). Our study identified oxygenated ROG species (OVOCs) as representative ROGs emitted from these sources, which are highly related to specific chemicals used during the industrial activities. Moreover, mass spectra similarity analysis revealed significant dissimilarities among the ROG emission sources, indicating substantial variations between different industrial VCP sources. Except for the ship coating industry utilizing solvent-borne coatings, the proportions of OVOCs range from 67% to 96% in total ROG emissions and 72% to 97% in total OH reactivity (OHR) for different industrial sources. The industrial VCP sources associated with solvent-borne coatings exhibited a higher ozone formation potential (OFP), reaching as high as 5.5 and 2.7 g $O_3 \cdot g^{-1}$ ROGs for ship coating and furniture coating industries, primarily due to contributions from aromatics. The fractions of the ten most abundant species in total ROG emissions, OHR, and OFP indicated a highly centralized of ROG emissions from various industrial VCP sources. Our results suggest that ROG treatment devices may have limited effectiveness for all ROGs, with treatment efficiencies ranging from -12% to 68%. Furthermore, we found that ROG pairs (e.g., methyl ethyl ketone (MEK) /$C_8$ aromatics ratio) could serve as effective indicators for distinguishing industrial VCP sources, particularly for measurements in industrial areas. Our study demonstrated the importance of measuring a large number of ROGs using PTR-ToF-



MS for characterizing ROG emissions from industrial VCP sources.



## 1. Introduction


With the successful control of vehicular emissions, emission from volatile
chemical products (VCPs) have become an increasingly significant source in cities all
around the world (Sun et al., 2018;McDonald et al., 2018;Li et al., 2019;Khare and
Gentner, 2018;Seltzer et al., 2022;Sasidharan et al., 2023). Reactive organic gases
(ROGs), organic gases other than methane, from VCPs emission can contribute
substantially to both anthropogenic secondary organic aerosol (SOA) and ozone ($O_3$)
in urban environments (Seltzer et al., 2022;Khare et al., 2022;Sasidharan et al.,
2023;Coggon et al., 2021;Gkatzelis et al., 2021b;Qin et al., 2021). With the
development of economy and industrialization, the emissions of industrial VCPs
contribute to approximately 25%-45% of ROG emissions in China (Ou et al., 2015;Wei
et al., 2011;Huang et al., 2011;Sha et al., 2021;Zhou et al., 2020b). To effectively
control atmospheric pollution in urban areas and surrounding regions, it becomes
imperative to gain a comprehensive understanding of the emission characteristics of
ROGs from industrial VCP sources.
Extensive research has been conducted to investigate ROG emissions from
industrial VCP sources, mainly focusing on sampling within manufacturing workshops
and exhaust stacks (Zheng et al., 2013;Yuan et al., 2010;Wang et al., 2014). Previous
studies have demonstrated that the use of individual chemicals (i.e. coatings, inks, and
adhesives) significantly impact on ROG emissions (Gkatzelis et al., 2021a;Zheng et al.,
2013;He et al., 2022a), and these chemicals used for printing, furniture, and shoemaking
industries has seen rapid growth and widespread adoption in recent years (Gkatzelis et
al., 2021a;McDonald et al., 2018;Seltzer et al., 2022;Coggon et al., 2021). Consequently,
the diverse emission sources and emission factors from industrial VCP sources have
contributed to large uncertainties (Mo et al., 2021;Zhong et al., 2018). To mitigate the
emissions of most primary pollutants, stricter emission standards have been
implemented along with advancements in ROG treatment technologies. Specifically,
water-borne VCPs has substituted solvent-borne VCPs in China (Mo et al., 2021;Li et
al., 2019;Shi et al., 2023). As a result, the emission characteristics of ROGs from



industrial VCP sources may undergo changes in response to the ongoing development
of VCPs and ROG treatment technologies. It is imperative to regularly updated the
understanding of ROG emission characteristics associated with industrial VCP sources.
The emissions of oxygenated ROG species (OVOCs) have been identified as
significant components in industrial VCP emissions (Chang et al., 2022;Mo et al.,
2021;Sha et al., 2021). For instance, it has been found that more than 80% of total ROG
emissions for shoemaking and printing industries are attributed to OVOC emissions
(Zheng et al., 2013). This notable contributions of OVOCs, such as acetone, methyl
ethyl ketone (MEK), ethyl acetate, and isopropanol, can be primarily attributed to the
use of individual industrial chemicals (Zheng et al., 2013;Wu et al., 2020b).
Traditionally, the collection of ROGs involved the use of canisters or Tedlar bags, and
their analysis was conducted using gas chromatography-mass spectrometer/flame
ionization detector (GC-MS/FID) techniques, with a primary focus on hydrocarbon
emissions (Yuan et al., 2010;Wang et al., 2014). Previous studies commonly employed
2,4-dinitrophenylhydrazine (DNPH) cartridges for collection and analyzed them using
high-performance liquid chromatography (HPLC) to detect carbonyl species such as
aldehydes and ketones. However, this approach is both time-consuming and susceptible
to contaminations (Mo et al., 2016;Han et al., 2019).
Due to the intricate chemical compositions of industrial VCPs, it is essential to
characterize ROG emissions with higher mass resolution. The proton-transfer-reaction
time-of-flight mass spectrometer (PTR-ToF-MS) has been extensively utilized for the
identification of VCP sources. It has been confirmed that VCP sources is a significant
contributor to ROG emissions. For instance, ROG emissions from VCP contribute 50%-
80% of anthropogenic ROG emissions in US cities (Gkatzelis et al., 2021b;McDonald
et al., 2018). The large fractions (~50%) of ROGs have been attributed to VCP-
dominated source in Guangzhou, highlighting its importance in urban environments (Li
et al., 2022). Through high mass resolution analysis, tracer compounds for various VCP
categories have been identified (Gkatzelis et al., 2021a;Coggon et al., 2018;Stockwell
et al., 2021). In addition, OVOCs such as acetates, acrylates, alcohols (e.g. benzyl



alcohol), glycols (e.g. propylene glycol, ethylene glycol), and glycol ethers, have been
found to make significant contributions to VCPs emission (Seltzer et al., 2021;Li and
Cocker, 2018;Li et al., 2018;Khare et al., 2022). With the ability to measure whole mass
spectra and offer high mass resolution, the PTR-ToF-MS enables more comprehensive
detection of a wide range of ROGs (Cappellin et al., 2012;Yuan et al., 2017;Huangfu et
al., 2021). By employing parameterization methods to determine instrument sensitivity,
more ROGs can be quantified from the obtained mass spectra (Sekimoto et al., 2017;Wu
et al., 2020a). Furthermore, previous studies have demonstrated that higher alkanes,
including acyclic, cyclic and bicyclic alkanes can be measured using PTR-ToF-MS with
$NO^+$ chemical ionization ($NO^+$ PTR-ToF-MS) (Inomata et al., 2014;Koss et al.,
2016;Wang et al., 2020;Chen et al., 2022). Higher alkanes are significant species in
vehicle and combustion emissions (Gao et al., 2023;Liu et al., 2021;Zhao et al., 2018b),
and they were not included in previous measurements of industrial VCP sources. Thus,
by combining hydrocarbons measured by offline GC-MS/FID, PTR-ToF-MS shows
promise as a method for developing more comprehensive speciation relevant to
industrial VCP emissions (Gao et al., 2023).
In this study, we applied a PTR-ToF-MS employing $H_3O^+$ and $NO^+$ chemical
ionization along with a GC-MS/FID to comprehensively measure ROG emissions from
five industrial VCP sources, including shoemaking, plastic surface coating, furniture
coating, printing, and ship coating industries in the Pearl River Delta (PRD) region of
China. We investigated emission characteristics of ROGs from semi-open workshops
and ROG treatment devices across these industries. We utilized the dataset to analyze
the contributions of different ROG components to total ROG emissions, OH reactivity
(OHR), ozone formation potential (OFP), and volatility in various industrial VCP
sources. Furthermore, we conducted intercomparisons of the mass spectra
characterizations of ROG emissions, which revealed significant variations in ROG
emissions from industrial VCP sources.
**2. Materials and methods**
**2.1 Tested industrial VCP sources and sampling methods**



Based on comprehensive analysis of written data, consultation with relevant
experts, and thorough on-site investigations, we selected five representative factories
and industries from various industrial VCP sources. The selection criteria for these
industries were based on relevant emission inventory research conducted in the PRD
region of China (Zhong et al., 2018). Sampling methods focused on capturing ROG
emissions generated during the main manufacturing processes, such as coatings
spraying and adhesives usage in the factories. Both online measurements and offline
sampling were carried out in semi-open workshops, as well as ROG treatment devices
(i.e. before and after emission treatment, generally located at the front and rear sampling
ports of the ROG treatment devices) in the factories (Table. S1).
During the campaign, a mobile monitoring vehicle was equipped with online
measurement equipment and strategically parked near the sampling ports of both
workshops and ROG treatment devices emissions (Fig. S1).    A $CO_2$ / $H_2O$ gas analyzer
(LI-COR 840A, Inc., USA) was used to measure the concentrations of $CO_2$ and $H_2O$.
To ensure continuous sampling, air from various factories was drawn through a length
of Perfluoroalkoxy (PFA) Teflon tubing, ranging from 10 to 100 meters, at a controlled
flow rate of 6 L/min facilitated by an external pump.
**2.2  ROG measurements**
In this study, ROG were measured using a proton transfer reaction quadrupole
interface time-of-flight mass spectrometer (PTR-QiToF-MS) (IONICON Analytik,
Innsbruck, Austria) (Sulzer et al., 2014) and a combination of canister sampling and
offline GC-MS/FID analysis system (canister-GC-MS/FID). More comprehensive
speciation of ROG was achieved by analyzing hydrocarbons by canister-GC-MS/FID,
quantifying all signals using $H_3O^+$ PTR-ToF-MS, and supplementing by acyclic, cyclic,
and bicyclic alkanes from $NO^+$ ionization of PTR-ToF-MS. The selection of
overlapping ROGs was similar to a previous study (Gao et al., 2023).
To capture the real-time emission characteristics of ROGs from industrial VCP
sources, the mass spectra of PTR-ToF-MS was recorded every 10 s. Prior to each test,
background measurements of the instrument were carried out by passing sampling air



through a custom-built platinum catalytical converter that had been preheated to 365 °C
for 1 minute. Throughout the campaign, the PTR-ToF-MS instrument automatically
alternated between two reagent ions ($H_3O^+$ and $NO^+$) every 10 minutes. Detailed setting
parameters for $H_3O^+$ and $NO^+$ chemical modes in this instrument can be found in
previous studies (Wu et al., 2020a;Wang et al., 2020;He et al., 2022b). The Tofware
software package (version 3.0.3, Tofwerk AG, Switzerland) was employed to facilitate
accurate data analysis (Stark et al., 2015).

Calibration for ROGs measure by PTR-ToF-MS were carried out both in the

laboratory and during the campaign. The PTR-ToF-MS was regularly calibrated using
a 23-component gas standard (Linde Spectra) throughout the campaign. During the later
period of the campaign, two gas standards (Apel Riemer Environmental Inc.) were used
for the calibration of other ROGs, specifically for acyclic and cyclic alkanes using $NO^+$
chemical ionization. (Wang et al., 2020;Chen et al., 2022;Wang et al., 2022). A total of
11 organic acids and nitrogen-containing compounds were calibrated using the liquid
calibration unit (LCU, IONICON Analytik, Innsbruck, Austria) (Table. S2-S4). In order
to account for the humidity dependence of some ROGs in the PTR-ToF-MS (Yuan et
al., 2017;Koss et al., 2018), humidity-dependence curves established in the laboratory
were utilized for correction (Wu et al., 2020a;He et al., 2022b;Wang et al., 2022).
Sensitivities of uncalibrated species were determined based on the kinetics of proton-
transfer reactions of $H_3O^+$ with ROGs (Fig. S2) (Cappellin et al., 2012;Sekimoto et al.,
2017), with an associated uncertainty of approximately 50% for the concentrations of
uncalibrated species.

Simultaneously, offline sampling was conducted near the sampling ports of

workshops and ROG treatment devices. Whole air samples were collected using
canisters for determination of hydrocarbons in industrial VCP sources, and analyzed by
an offline GC-MS/FID system.. The GC-MS/FID system was calibrated using
photochemical assessment monitoring stations (PAMS) and TO-15 standard mixtures,
which enabled the identification and quantification of a total of 94 hydrocarbons. More
information about this instrument and dataset for canister sampling and offline GC-



MS/FID system can be found elsewhere (Li et al., 2020).
**2.3 Calibrations of esters and isopropanol based on $H_3O^+$ and $NO^+$**

**ionization**

Since ester species (including acetates and acrylates) play a significant role in

industrial VCP sources, it is important to accurately quantify their concentrations
(Khare et al., 2022). Previous studies have demonstrated that ethyl acetate exhibits
notable fragmentation, resulting in interference at m/z 61 (e.g. $C_2H_4O_2H^+$) and m/z 43
(e.g. $C_2H_2OH^+$) (Haase et al., 2012;de Gouw and Warneke, 2007;Rogers et al.,
2006;Fortner et al., 2009). Therefore, we employed the PTR-ToF-MS to directly
measure high-purity ester chemicals and identify the characteristic product ions
produced by esters under $H_3O^+$ and $NO^+$ chemical ionization. Several common esters
including methyl acetate, ethyl acetate, isopropyl acetate, and vinyl acetate, were
selected to investigate instrument fragmentation under different ionizations. As shown
in Table. S5, it is intriguing to observe that high-molecular-weight acetates tent to
exhibit more fragmentation, resulting in interference at m/z 61 (e.g. $C_2H_4O_2H^+$) and m/z
43 (e.g. $C_2H_2OH^+$). Methyl acetate (95%) and ethyl acetate (72%) displayed limited
fragmentation in the instrument, while isopropyl acetate accounted for only 13% of the
$C_5H_{10}O_2H^+$ ions. Additionally, esters with different chemical structures may undergo
distinct modes of fragmentation. For example, vinyl acetate primarily fragmented to
produce interfering fragments at m/z 43 (e.g. $C_2H_2OH^+$) with a fraction of 78%.
Furthermore, considering the PTR-ToF-MS mass spectra from various industrial VCP
sources, it is conceivable that other ester compounds might also contribute to these mass
channels, emphasizing the need for cautious consideration of m/z 61 (e.g. $C_2H_4O_2H^+$)
and m/z 43 (e.g. $C_2H_2OH^+$) signals measured by $H_3O^+$ PTR-ToF-MS in industrial VCP
sources. The use of $NO^+$ chemical ionization exhibits various reaction pathways with
ROGs (Wang et al., 2020;Chen et al., 2022), which can partially mitigate interference
from fragment ions (Table. S5). The identified results of acetates based on $NO^+$
ionization demonstrated considerable improvements for methyl acetate (83%) and ethyl
acetate (80%), whereas vinyl acetate exhibited more fragmentation, with the largest





contribution (47%) at m/z 43 (e.g. $C_2H_2OH^+$). This result could be explained by the
instrument was more likely to have a fracture reaction due to the chemical structure of
vinyl acetate, which contains a C=C bond.

Additionally, it is challenging to calibrate isopropanol in the $H_3O^+$ PTR-ToF-MS

since alcohols split off water during ionization (Buhr et al., 2002). To overcome this
challenge, we implemented daily calibrations of isopropanol under ambient humidity
conditions throughout the campaign (Fig. S3). The $NO^+$ PTR-ToF-MS was also
employed to calibrate isopropanol by identifying the characteristic product ions
produced under $NO^+$ ionization (Table. S5). The dominating product ion of isopropanol
was observed at m/z 59 (e.g. $C_3H_7O^+$) (88%), which corresponds to acetone ($C_3H_6OH^+$)
ions in the $H_3O^+$ PTR-ToF-MS. Although the dominant product ion for acetone under
$NO^+$ ionization was observed at m/z 88 (e.g. $C_3H_6O(NO)^+$) (77%), the interfere at m/z
59 (e.g.$C_3H_6OH^+$) (23%) was not insignificant. Therefore, the concentration of
isopropanol measured by $NO^+$ PTR-ToF-MS in this campaign has eliminated the
influence of acetone. Finally, the good agreement between measurements obtained
using PTR-ToF-MS with $H_3O^+$ and $NO^+$ chemical ionization throughout the campaign
indicates that the $NO^+$ PTR-ToF-MS can serve as a reliable method for measuring
isopropanol and ester species in industrial VCP sources (Fig. S4-S5). Our results
demonstrated that the $NO^+$ PTR-ToF-MS can also provide a complementary approach
for characterizing ester species and isopropanol in ambient air as well as emission
sources.

### 2.4  Mass spectra similarity analysis

We conducted a comprehensive comparison of various ROG emission sources by

considering the entire range of species in mass spectra as dimensions in a vector, and
calculating the cosine angle ($\theta$) similarity (Humes et al., 2022;Ulbrich et al.,
2009;Kostenidou et al., 2009). The angle $\theta$ between the two mass spectra ($MS_a$ and $MS_b$)
is given by the following:
$$cos\,\theta = \frac{MS_a MS_b}{|MS_a||MS_b|} \tag{1}$$

The $\theta$ angles between two mass spectra is divided into 4 groups, including 0°-



15°, 15°-30°, 30°-50°, and>50°, which correspond to excellent consistency, good
consistency, many similarities, and poor consistency, respectively. Due to the distinct
ionization methods of the instruments, our classification of angle similarity is not as
strict as that reported in previous studies (Kostenidou et al., 2009;Zhu et al., 2021),

## 3. Results and discussions

### 3.1 Time-resolved ROG emissions from industrial VCP sources

Time series of several ROGs measured by the $H_3O^+$ PTR-ToF-MS from five
industrial VCP sources are shown in Fig. 1 and Fig. S6. More information for these
sources can be found in Sect. S1 in the Supplement. Online measurements were carried
out in semi-open workshops (workshops emission) and from ROG treatment devices
(i.e. before and after treatment emission). Typically, workshop waste gases are routed
through collection devices, followed by collection and treatment in ROG treatment
devices, before being released into the atmosphere via exhaust stacks. ROG treatment
devices are implemented to reduce ROG emissions after treatment, thereby ensuring
that the ROG concentrations after treatment are generally lower than those before
treatment. As the waste gas was directly discharged into the ambient after treatment,
the after treatment emission was considered as stack emission (Zheng et al., 2013). The
average concentrations of eight representative ROGs, including aromatics, ketones,
alcohols, and esters, between workshops emission and stack emission for all factories
is presented in Fig. S7. The evaluation of the ROGs treatment efficiency is based on the
analysis of emission characteristics before and after treatment in the ROG treatment
devices, which is discussed in greater detail in Section 3.3. Along with the typical ROGs,
the PTR-ToF-MS measured a wide range of ions in abundance in the mass spectra. Fig.
2 displays mass spectra representing the average concentrations of stack emissions from
five industrial VCP sources for all detected ROGs. These ROGs measured by the PTR-
ToF-MS were categorized based on their chemical formula, namely hydrocarbon
species ($C_xH_y$), OVOCs ($C_xH_yO_1$, $C_xH_yO_2$, and $C_xH_yO_{\geq 3}$), species containing nitrogen
and/or sulfur atoms (N/S-containing), species containing siloxanes (Si-containing), and
other ions (others).



Real-time concentrations of toluene, acetone, ethyl acetate, and isopropanol from

the shoemaking industry are displayed in Fig. 1a. The variable manufacturing processes
conditions are demonstrated by inconsistent emission levels in the workshops. This
variation may be attributed to different emission intensities during different periods.
Notably, the significant emissions from the shoemaking industry are primarily
attributed to a few low-molecular-weight OVOCs (Fig. 2a), including acetone, MEK,
isopropanol, and formaldehyde, followed by a fraction of hydrocarbon species (e.g.
toluene). Our results align with previous findings (Zheng et al., 2013;Zhao et al., 2018a),
emphasizing that raw chemicals used during the industrial activities play crucial roles
in determining the constituents of the industrial VCP emissions.

Significant variations in ROG concentrations were also observed from the plastic

surface coating industry (Fig 1b). These variations could be attributed to different
manufacturing process conditions and the use of different chemicals in workshops as
well. As shown in Fig. 2b, OVOCs contribute significantly to emissions from this
industry. Representative OVOCs for $C_xH_yO_1$ ions consist of isopropanol, acetone,
formaldehyde, methanol, and cyclohexanone. $C_xH_yO_2$ ions refer to acetates and
acrylates such as $C_3H_6O_2$ (e.g. methyl acetate), $C_6H_{12}O_2$ (e.g. butyl acetate), $C_9H_{16}O_2$
(e.g. allyl hexanoate) and $C_{12}H_{20}O_2$ (e.g. linalyl acetate). Notably, there are some
differences from the main components compared to previous results (Zhong et al.,
2017), which may be attributed to the substitution of solvent-borne chemicals with
water-borne chemicals in industrial VCPs. Moreover, the utilization of PTR-ToF-MS
enabled the identification of additional important OVOCs, thereby improving the
characterization of ROG emissions from the industrial VCPs.

Due to the wide variety of industrial coatings used in the furniture coating

industry, numerous ROGs can be observed in the measured mass spectra (Fig. 2c).
Notably, $C_xH_yO_2$ (24%) and $C_xH_yO_3$ ions (9%) contribute significantly in this industry.
Among the identified species, $C_8$ aromatics exhibit the highest concentrations,
consistent with previous research from industries utilizing solvent-borne coatings (Yuan
et al., 2010;Wu et al., 2020b;Wang et al., 2014). Other OVOCs such as MEK, ethanol,



and formaldehyde for $C_xH_yO_1$ ions, $C_6H_{12}O_2$ (e.g. butyl acetate), $C_5H_8O_2$ (e.g. methyl
methacrylate, acetylacetone) for $C_xH_yO_2$ ions, and $C_6H_{12}O_3$ (e.g. propylene glycol
methyl ether acetate, PGMEA) and $C_7H_{14}O_3$ (e.g. butyl lactate) for $C_xH_yO_3$ ions had
been found may be associated with emissions from water-borne coatings. This finding
underscores the importance of considering high-molecular-weight OVOCs in this
industry, further emphasizing the ability of PTR-ToF-MS to better characterize these
important OVOCs that serve as raw chemicals for industrial VCPs.

Moreover, by employing online PTR-ToF-MS technology, we can gain deeper

insights into the emission characteristics of ROGs during both working and non-
working hours. We conducted an analysis of ROG emissions in the furniture coating
factory during non-working hours (from 10:00 p.m. to 8:00 a.m. the next day) and
compared them with emissions during working hours (Fig. 1c). Most ROGs exhibited
a gradual decrease in concentration during non-working hours, with the exception of
formaldehyde which maintained a constant concentration. Notably, the concentrations
of other typical ROGs, such as MEK and $C_8$ aromatics, were 2-5 times lower during
non-working hours compared to working hours. This observation suggests that ROGs
may still be emitted even when the painting activities in the factory is halted, with night-
time emissions accounting for approximately 20% of total daily emissions. The $\theta$ angles
of mass spectra between real-time concentrations versus working hours shows that
ROG emissions are many similarities during both working and non-working hours (Fig.
1d, $\theta<30°$ in most times). Given that some ROGs were still more abundant and
continued to be released into the atmosphere even during non-working hours (e.g. from
the volatilization of chemicals), the ROG emissions in factories during non-working
hours should not be ignored.

The real-time concentrations of typical ROGs measured from the printing

industry is shown in Fig. S6a, with an emphasize on the performance of two different
ROG treatment devices, namely activated carbon adsorption combined with ultraviolet-
ray (UV) photolysis devices and catalytic combustion devices (specifically,
regenerative thermal oxidizer (RTO) devices) installed in this factory. Isopropanol was



found to have the highest concentration in the printing industry (Fig. 2d), which is
consistent with previous studies (Zheng et al., 2013). The higher concentrations of other
typical species, such as $C_4H_8O_2$ (e.g. ethyl acetate), $C_5H_{10}O_2$ (e.g. isopropyl acetate),
and $C_7H_{16}O_3$ (e.g. dipropylene glycol methyl ether, DPM) substantiate the correlation
between ROG emissions and industrial inks utilized in the printing industry. It was
found that ROG treatment devices exhibit varying treatment efficiencies for ROGs,
particularly for OVOCs (such as isopropanol and ethanol), that may not have been
effectively removed by these treatment devices.
In comparison to other industrial VCP sources, the ship coating industry exhibits
the highest emissions of hydrocarbons (86%), specifically $C_6$-$C_{11}$ aromatics (Fig. 2e,
also in Fig. S6b, Sect. S1). This may be attributed to the utilization of solvent-borne
industrial coatings for ship coating remains prevalent due to stringent requirements for
anti-rust and anti-corrosion properties (Malherbe and Mandin, 2007). A few OVOCs,
such as methanol and MEK, were identified as significant emissions. These results
confirm that ROG emissions from solvent-borne coatings, predominantly composed of
$C_8$ aromatics, continue to be the primary contributors in the ship coating industry, which
is consistent with a previous study conducted in the PRD region (Zhong et al., 2017).
Fig. 2 provides a quantified of the proportions of different ion categories
measured by the PTR-ToF-MS across various industrial VCP sources as well. OVOCs
make up the largest fractions in the printing (94%), plastic surface coating (90%),
shoemaking (84%), and furniture coating (68%) industries, while they only account for
13% of emissions from the ship coating industry. The fractions of different OVOC
groups exhibit a general decline    from $C_xH_yO_1$ to $C_xH_yO_{\geq3}$, and OVOCs with more
than two oxygen atoms are present in small proportions (0.3%-8.5%) in all industrial
VCP sources except for the furniture coating industry (33%), indicating little emissions
of these species. However, although these OVOCs with two or more oxygen atoms do
not contribute significantly to the overall emissions, some of them may serve as tracer
compounds for particular emission sources as they were only detected in single source.
Previous studies have identified octamethylcyclotetrasiloxane (D$_4$ siloxane), texanol





(C$_{12}$H$_{24}$O$_3$) and para-chlorobenzotrifluoride (PCBTF, C$_7$H$_4$ClF$_3$) as tracer compounds
for individual chemicals (adhesives and coatings) in U.S. cities (Gkatzelis et al., 2021a).
We also observed that the concentrations of texanol and PCBTF emitted by relevant
industrial VCP sources were unique and almost non-existent in other sources. Texanol
was only detected in samples from the plastic surface coating and furniture coating
industries that utilize water-borne coatings. Similarly, PCBTF was only found in
samples from the ship coating and furniture coating industries that use solvent-borne
coatings. These findings suggest that texanol and PCBTF may be applicable as tracer
compounds for industrial VCPs in China. On the contrary, D$_4$ siloxane was not found
to be specific to emissions from adhesive-related industrial (i.e. shoemaking industry)
(Fig. 1), indicating that D$_4$ siloxane may not be an appropriate tracer compounds for
identifying industrial VCPs in China.
**3.2 Distributions of ROG emissions, OHR, OFP, and volatility**

We compared the mass spectra of these industrial VCP sources and calculating

the $\theta$ angles similarity (Fig. 3) (Table. S6). The ROGs showed a diverse similarity
among different types of industrial VCP sources. Only plastic surface coating industry
versus printing industry demonstrated good consistency (27°), other mass spectra
exhibited poor consistency ($\theta$>60°). Combined with mass spectra of vehicular
emissions (Wang et al., 2022), the $\theta$ angle similarities among the mass spectra of
industrial VCP sources (62°-90°) were worse than those of vehicular emissions (41°-
75°) (Fig. 3). It is interesting to observe that the $\theta$ angle similarity among the mass
spectra in different workshops in printing and ship coating industries ranged from1.6°
to 9.0° (Table. S7), similar to the mass spectra in various emission standards for
gasoline vehicles (4.9°-17°) (Table. S8). Conversely, the $\theta$ angle similarity among the
mass spectra of workshops in other industrial VCP sources ranged from 13° to 60°,
indicating significant differences in ROG emissions from industrial VCP sources. These
substantial differences indicate that ROG emissions from industrial VCPs are more
complex and diverse than vehicular emissions. Consequently, a more accurate
classification of industrial VCP emissions is necessary, as they cannot be directly



unified as a single class of emission sources.

The combination of PTR-ToF-MS and canister-GC-MS/FID measurements

allowed for more comprehensive speciation of ROG emissions from industrial VCP
sources. This comprehensive approach enabled the determination of the fractions of
ROGs in total ROG emissions for various industrial VCP sources (Table. S9, Fig. S5,
details in Sect. S2 in the Supplement). Additionally, ROGs reactivity plays a crucial
role in characterizing the contributions of different ROGs to atmospheric chemical
reactions and the formation of secondary pollutants (Wu et al., 2020a;Yang et al., 2016).
The overall OHR of ROGs was calculated to comprehend the role of ROGs emitted by
industrial VCP sources. The calculation only employed ROGs with known reaction rate
constants with the OH radical, which were taken from previous studies (Atkinson and
Arey, 2003;Atkinson et al., 2004;Atkinson et al., 2006;Koss et al., 2018;Wu et al.,
2020a;Zhao et al., 2016). The fractions of ROGs in the total OHR of ROGs can be
determined for various industrial VCP sources (Table. S10). ROGs are grouped into
categories, including OVOCs, N/S-containing, and heavy aromatics and monoterpenes
measured by $H_3O^+$ PTR-ToF-MS, higher alkanes (including $C_{10}$-$C_{20}$ acyclic, cyclic, and
bicyclic cycloalkanes) measured by $NO^+$ PTR-ToF-MS, and alkanes, alkenes, aromatics,
and halohydrocarbons measured by canister-GC-MS/FID.

OVOCs contributed significantly to total ROG emissions (Fig. 4a), and fractions

of OVOCs in total ROG emissions are comparable to previous studies (Fig. 5). Notably,
OVOCs account for 67% of total ROG emissions from the shoemaking industry, which
is slightly lower than findings from other studies in the PRD region (Zheng et al., 2013)
but higher than those reported in previous studies (Zhou et al., 2020a;Zhao et al., 2018a).
The fractions of OVOCs in total ROG emissions from plastic surface coating, printing,
and furniture coating industries are 96%±0.2%, 85%±6.5%, and 77%, respectively.
Compared to previous studies (Zhong et al., 2017;Zheng et al., 2013;Fang et al.,
2019;Zhao et al., 2018a;Wang et al., 2019;Zhou et al., 2020a;Zhao et al., 2021),
determined OVOC fractions for these industrial VCP sources are much higher (Fig. 5),
which may be related to two reasons: (1) more OVOC species are detected in this study;





(2) water-borne coatings and inks are more widely employed in the recent year which
may enhance OVOC fractions. Moreover, OVOCs account for 16%±3.5% of total ROG
emissions from the ship coating industry by using the solvent-borne coatings, and the
fraction is also higher than findings from in one previous study (Zhong et al., 2017).
Additionally, OVOCs also contribute to the largest fraction in total OHR of ROGs from
all industrial VCP sources (72%-97%) except for the ship coating industry (15%±3.6%)
(Fig. 4b). In contrast to the important contribution of OVOCs, the fractions of
hydrocarbons measured by canister-GC-MS/FID only made considerable contributions
in specific industrial VCP sources (Fig. 4). For instance, aromatics were found to be the
major contributors to both total ROG emissions and OHR in the ship coating industry,
making up 74%±6.1% and 79%±4.8% respectively. Alkanes measured by canister-GC-
MS/FID only make contributions in the shoemaking industry, comprising 26% of the
total ROG emissions. Overall, the total OHR of ROGs was dominated by OVOCs and
aromatics, and the contributions of other species were in the range of 1.8%-21% (Fig.
4b). These results stress the importance of measuring a broad range of OVOCs using
PTR-ToF-MS in characterizing ROG emissions from industrial VCP sources.
The application of NO$^+$ PTR-ToF-MS provided the opportunity for detecting
emissions of higher alkanes from industrial VCP sources. We show that the contribution
of higher alkanes can be significant for VCP sources. Specifically, the printing industry
demonstrates a noteworthy presence of higher alkanes, accounting for 27%±2.7% and
8.2%±2.4% in workshop and stack emissions, respectively (Table. S9). This can be
attributed to the use of lubricating oil, a primary component of industrial inks, which
contains substantial amounts of alkanes (Liang et al., 2018). Furthermore, emissions
from forklifts transporting products in printing workshops also contribute to the
emission of higher alkanes (Li et al., 2021), suggesting non-road vehicles may
contribute to the emissions from industrial VCP factories. In addition, the fractions of
higher alkanes in stack emission are lower than in workshops, suggesting that ROG
treatment devices effectively reduce emissions of higher alkanes.
To facilitate for making controlling strategies for ozone, we determine the ozone



formation potential for a unity of emission from different sources for comparison (Yuan
et al., 2010;Na and Pyo Kim, 2007), which represent the ability to ozone formation
from ROG sources on a relative basis (Fig. 6) Our calculations specifically focused on
ROGs with known maximum incremental reactivity (MIR) values, derived from
previous studies (Carter, 2007). Among the industrial VCP sources considered, the ship
coating industry exhibited the highest OFP, reaching as high as 5.5 g $O_3 \cdot g^{-1}$ ROGs,
followed by the furniture coating industry, with a value of 2.7 g $O_3 \cdot g^{-1}$ ROGs. The OFP
for other industrial VCP sources ranged from 0.79 g $O_3 \cdot g^{-1}$ ROGs to1.4 g $O_3 \cdot g^{-1}$ ROGs.
Among all industrial VCP sources, aromatics (ranging from 4.2% to 91%) and OVOCs
(ranging from 6.7% to 94%) were identified as the primary contributors to OFP.
Compared to vehicular emissions, the OFP from the ship coating and furniture coating
industries are significantly higher (Fig. 5), suggesting that these sources should be
controlled in priority. Given the higher reactivity value for ship coating industry relative
to other sources, it is evident that controlling ROG emissions from solvent-borne
industrial chemicals would have a more substantial impact on reducing ozone formation
compared to other sources. Moreover, it is important to note that the emissions of
solvent-borne chemicals surpass those of vehicles, while water-borne chemicals have
lower emissions compared to vehicles. This observation implies that the substitution of
solvent-borne chemicals with water-borne chemicals in China holds considerable
importance in mitigating and controlling ozone pollution.
We further compare centralization for species among different ROGs sources by
determining the contribution from the top ten species in terms of concentrations, OHR,
and OFP (Fig. 7 and Fig S8, also in Table S11). We show that the top ten ROGs account
for over 50% on ROG emissions, OHR, and OFP (Fig. 7). With the exception of
furniture coating industry, the fractions on the top ten species in total emissions, OHR,
and OFP from industrial VCP sources were in range of 89%-96%. The lower fractions
(ranging from 69% to 86%) of the top ten species in the furniture coating industry may
be a result of the wider range of industrial coatings (i.e. both solvent-borne and water-
borne coatings) utilized in this industry. ROGs emitted from industrial VCP sources are



apparently more centralized compared to vehicular emissions (ranging from 51% to 87%). Additionally, the cumulative fractions of the top one hundred species in overall ROG emissions, OHR, and OFP in various industrial VCP sources is further indicated the highly centralized of ROG emissions from various emission sources (Fig. S8). More than half of the top ten species in ROG emissions, OHR, and OFP from industrial VCP sources were OVOCs (Table S11). Among them, isopropanol made a notable contribution to the printing, plastic surface coating, and shoemaking industries. Other OVOCs such as MEK, acetone, and ethyl acetate contributed to total ROG emissions in each industry, while formaldehyde, acetaldehyde contributed to total OHR and OFP. It should be noted that the proportions of $C_{13}$, $C_{14}$, and $C_{15}$ cycloalkanes from printing industry (account for 6.3% in ROG emissions), as well as the proportion of $C_{11}$ aromatics from ship coating industry (account for 1.0% in ROG emissions) are not negligible. Additionally, acetylacetone is a common species with broad industrial applications, and contributes importantly to secondary pollutants formation under polluted environments (Ji et al., 2018). Although it only contributes 8.7% to total emissions from the furniture coating industry, its fraction in terms of total OHR can be as high as 30%. These findings demonstrated that previously underreported ROGs should receive greater attention in future research.

The updated measurements of OVOC emissions by using the PTR-ToF-MS substantially improve our understanding of the emission of industrial VCP sources. The effective saturation concentrations (C*) of high-molecular-weight OVOCs were found to be lower, which were corresponding to intermediate-volatility organic compounds (IVOCs) and semi-volatile organic compounds (SVOCs). Since these S/I-VOCs are crucial precursors for the SOA in urban environments (Zhao et al., 2014), it is important to comprehend their contributions from the emissions of industrial VCP sources across various volatility classes, including volatility organic compounds (VOCs), IVOCs, and SVOCs (Guenther et al., 2012;Li et al., 2016). Fig. 8 illustrates the distribution of ROG species in a two-dimensional volatility basis set (2D-VBS) space for various industrial VCP sources, categorized based on volatility bins (Li et al., 2016;Donahue et al., 2011).



It is worth noting that the volatility distributions exhibit substantial variation across
industrial VCP sources (Fig. 8a). Generally, VOCs constitute the predominant fraction
of emissions from industrial VCP sources, accounting for 59% to 98% of the total
emissions. The fractions of IVOCs are largest in the printing industry (40%), compared
to the range of 2.1%-9.6% in other industrial VCP sources. Conversely, the contribution
of SVOCs from industrial VCP sources are negligible in our study, accounting for less
than 1%. Considering the importance of S/I-VOCs in SOA formation, particularly with
the increasing adoption of improved online mass spectrometry technologies, the S/I-
VOCs emissions from industrial VCP sources should be paid more attention in future
research.

### 3.3  Evaluate ROGs treatment efficiency in industrial VCP sources


The analysis of the PTR-ToF-MS mass spectra offers valuable insights into the
impact of ROG emissions from industrial VCP sources. This comprehensive
information provided by the PTR-ToF-MS also allows for a systematic comparison of
emissions before and after the treatment of ROGs. ROG treatment devices are
employed to reduce ROG emissions after treatment. Here, we evaluate two types of
ROG treatment devices: activated carbon adsorption combined with UV photolysis
devices (installed in shoemaking, plastic surface coating, furniture coating, and printing
industries) and catalytic combustion devices (installed in printing and ship coating
industries).
The scatterplot of the concentrations of various ROG before and after treatment
in industrial VCP sources are shown in Fig. 9. The observed treatment efficiency,
represented by 1-slope, did not reach the desired levels, ranging from -12% to 68%.
Among the industrial VCP sources investigated, the shoemaking industry exhibited the
highest treatment efficiency (slope=0.32) with the activated carbon adsorption
combined with UV photolysis device. This remarkable efficiency can be attributed to
the large-scale nature of the factory and meticulous regulation of the ROG treatment
devices. Following closely behind is the printing industry, utilizing catalytic
combustion devices, with a slightly higher efficacy (slope=0.67) than another treatment





device in the same factory (slope=0.80). However, it is evident that the treatment
efficiency has not reached the desired levels, possibly due to the challenges associated
with effectively removing OVOC emissions from the printing industry using current
treatment technologies. Additionally, we also observed that some OVOCs may be
generated as byproducts after the implementation of treatment devices in printing
industry. For instance, the concentrations of $CH_2O_2$ (e.g. formic acids), $C_4H_6O_3$ (e.g.
propylene carbonate), and $C_9H_{18}O$ (e.g. nonanal) were found to be higher after the
application of activated carbon adsorption combined with UV photolysis devices (Fig.
9d). Similarly, the concentrations of $C_3H_4O$ (e.g. acrolein) and $C_{12}H_{18}O_4$ (e.g. dibutyl
squarate) were also higher following the utilization of catalytic combustion devices (Fig.
9e). Therefore, it is crucial to consider the potential contribution of these ROGs when
assessing the emissions released into the atmosphere. The lowest treatment efficiency
of ROG was shown in the furniture coating industry (slope=1.12). This potentially
attributed to the ineffective performance of the ROG treatment devices in this particular
facility, as activated carbon and other adsorption materials were not promptly replaced.

Furthermore, the $\theta$ angles between the mass spectra of ROG from workshops,

before and after ROG treatment devices for various industrial VCP sources were
calculated and summarized in Fig. S9 (also in Table. S12). A comparison between the
correlation of mass spectra among workshops versus after treatment devices (ranging
from 6.2° to 49°) and workshops versus before treatment devices (ranging from 4.2° to
41°) demonstrated a poorer correlation in the former case. The similarities between
workshops and stack emissions in the shoemaking industry were lower compared to
other industrial VCP sources. This discrepancy can potentially be attributed to the
inclusion of ROG emissions from non-VCP usage manufacturing processes (e.g. sole
injection molding) in the collection process of ROG treatment devices. Additionally,
the $\theta$ angles similarity between the mass spectra from before and after ROG treatment
devices in various industrial VCP sources also providing insight into the efficacy of the
devices in removing ROGs (Fig. 9). The $\theta$ angles in ROG treatment devices from five
industrial VCP sources were found to range from 1.8° to 27°, indicating good





consistency between the mass spectra before and after treatment of ROGs. This
alignment suggests that the chemical compositions of ROG emissions remain
comparable before and after treatment (R ≥ 0.87), implying that the relative proportions
of various ROG components are not significantly affected by the ROG treatment
devices in these industrial VCP sources.

### 3.4 Comparison of industrial VCP sources and ambient air

To gain deeper insights into the atmospheric impact of emissions from industrial
VCP sources, an in-situ measurement was carried out at a monitoring station nearby the
furniture coating industry, located 2 km northeast from the industry site. The
measurement was conducted using a PTR-ToF-MS (Kore Inc., U.K.), which enabled
the quantification of various common ROGs. More information about this instrument
and dataset for in-situ measurement can be found elsewhere (Gonzalez-Mendez et al.,
2016;Song et al., 2023). Concordant with expectations, the average concentrations of
representative ROGs generally demonstrate a discernible decline from the furniture
coating industry (including stack emission and workshops during working and non-
working hours) to the monitor station (Fig. 10). $C_8$ aromatics and MEK performed
considerable emissions from furniture coating industry, while the concentrations of
them in the ambient air are orders of magnitude lower than those observed in the
industry. However, ambient concentrations of OVOCs, specifically MEK (6.8±8.2 ppb)
and ethyl acetate (7.5±5.9 ppb) are still significantly higher than clean environments
and are among the highest measured concentration in the literature (Wu et al., 2020a;He
et al., 2022b;Khare et al., 2022;Yang et al., 2022). It is confirmed that OVOCs should
be paid attention to in industrialized urban areas, thereby further substantiating the
significance of OVOC emissions from industrial VCP sources to atmospheric pollution.
These results stress the invaluable insights provided by PTR-ToF-MS in
comprehensively characterizing ROG compositions in both emission sources and urban
air.
The preceding discussions illustrate that the emission characteristics of ROGs
significantly vary among industrial VCP sources. As a result, the ratio of ROG pairs



can be used to distinguish emissions of industrial VCP sources. MEK and $C_8$ aromatics
emerge as key species in industrial VCP emissions, and the reaction rate constants of
$C_8$ aromatics with OH radical ($k_{OH}$ = (1.4-2.3) × $10^{-11}$ $cm^3 \cdot molecule^{-1} \cdot s^{-1}$) are higher
than MEK ($k_{OH}$ = 5.5 × $10^{-12}$ $cm^3 \cdot molecule^{-1} \cdot s^{-1}$) (Atkinson and Arey, 2003;Wu et al.,
2020a). Fig. 11 illustrates the correlation of MEK with $C_8$ aromatics in stack emission,
workshops during working hours and non-working hours in the furniture coating
industry, as well as ambient measurement near the industry. Positive correlations
between MEK and $C_8$ aromatics are observed in both emission sources and ambient
measurements, indicating a shared source for these compounds. Additionally, the
observed ratios of MEK to $C_8$ aromatics in ambient measurement are also comparable
with the ratios of MEK to $C_8$ aromatics measured in emissions from the furniture
coating industry (0.97 $ppb \cdot ppb^{-1}$ for the stack emission and 0.75 $ppb \cdot ppb^{-1}$ for the
workshops emission), suggesting that industrial VCP emissions (specifically furniture
coating) may account for the enhancement of MEK and C8 aromatics in this industrial
area. Thus, the divergence in MEK / $C_8$ aromatics ratio among different industrial VCP
sources suggests that these ratios could serve as effective indicators for distinguishing
industrial VCP emissions, particularly in ambient measurements within industrial areas.
Consequently, the MEK to $C_8$ aromatics ratio could provide an additional tracer for
assessing the contribution of industrial VCP emissions by using high time-resolution
ROG measurements from PTR-ToF-MS.

## 4. Conclusion

In this work, we conducted a field campaign to measure more comprehensive

speciation of ROG emissions from industrial VCP sources, including shoemaking,
plastic surface coating, furniture coating, printing, and ship coating industries. To
achieve this, we employed the PTR-ToF-MS in combination with canister-GC-MS/FID
techniques. Our study identified OVOCs had been identified as representative ROGs
emitted from these sources, which are highly related to specific chemicals used during
the industrial activities. Furthermore, we performed a mass spectra similarity analysis
to compare the ROG emissions across different emission sources. The poor consistency





of the similarity between the mass spectra in emission sources indicating that substantial
differences between industrial VCP sources, as they cannot be directly categorized as a
single class of emission sources.
In addition, the fractions of ROGs in total ROG emissions and OHR are
determined by combining measurements from canister-GC-MS/FID and PTR-ToF-MS.
Except for the ship coating industry utilizing solvent-borne coatings, the proportions of
OVOCs range from 67% to 96% in total ROG emissions and 72% to 97% in total OHR
for different industrial sources. Large fraction of OVOCs may be related to two reasons:
(1) more OVOC species are detected in this study; (2) water-borne coatings and inks
are more widely employed in the recent year which may enhance OVOC fractions. This
highlights the importance of measuring these OVOC emissions from industrial VCP
sources. The industrial VCP sources associated with solvent-borne coatings exhibited a
higher OFP, reaching as high as 5.5 and 2.7 g $O_3 \cdot g^{-1}$ ROGs for ship coating and
furniture coating industries, primarily due to contributions from aromatics, suggesting
that these sources should be controlled in priority. The fractions of the ten most
abundant species in total ROG emissions, OHR, and OFP indicate the highly centralized
of ROG emissions from various emission sources.
Our results suggest that ROG treatment devices may have limited effectiveness
in removing ROGs, with treatment efficiencies ranging from -12% to 68%. In addition,
the average concentrations of representative ROGs generally demonstrate a downward
trend from emission sources to the ambient air. Our results  demonstrate that ROG
pairs (e.g., MEK / $C_8$ aromatics ratio) can be utilized as reliable indicators for
distinguishing industrial VCP sources, particularly for measurements in industrial areas.
This study highlights the significant role of OVOCs to ROG emissions from
industrial VCP sources, particularly those utilizing water-borne chemicals. As a result,
these industrial VCP sources may significantly contribute to the primary emissions of
OVOCs in urban regions. The current emission inventories do not fully account for the
emissions of many ROGs, which can compromise the predictive accuracy of air quality
models in urban areas. In this study, a broader range of ROG species was quantified



using PTR-ToF-MS measurements, which highlights the effectiveness of PTR-ToF-MS
in characterizing ROG emissions from industrial VCP sources..

## Data availability

Data are available from the authors upon request.

## Author contribution

BY designed the research. BY and SHW organized industrial VCP source
measurements. SHW, XJH, RC, CHW, and CMW contributed to data collection. SHW
performed the data analysis, with contributions from XS and YBC. SHW and BY
prepared the manuscript with contributions from YBH, XBL, BGW and MS. All the
authors reviewed the manuscript.

## Competing interests

The authors declare that they have no known competing financial interests or
personal relationships that could have appeared to influence the work reported in this
paper.

## Acknowledgement

This work was supported by the National Key R&D Plan of China (grant No.
2022YFC3700604), the National Natural Science Foundation of China (grant No.
42230701). This work was also Supported by the Outstanding Innovative Talents
Cultivation Funded Programs for Doctoral Students of Jinan University (grant
No.2022CXB028).



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

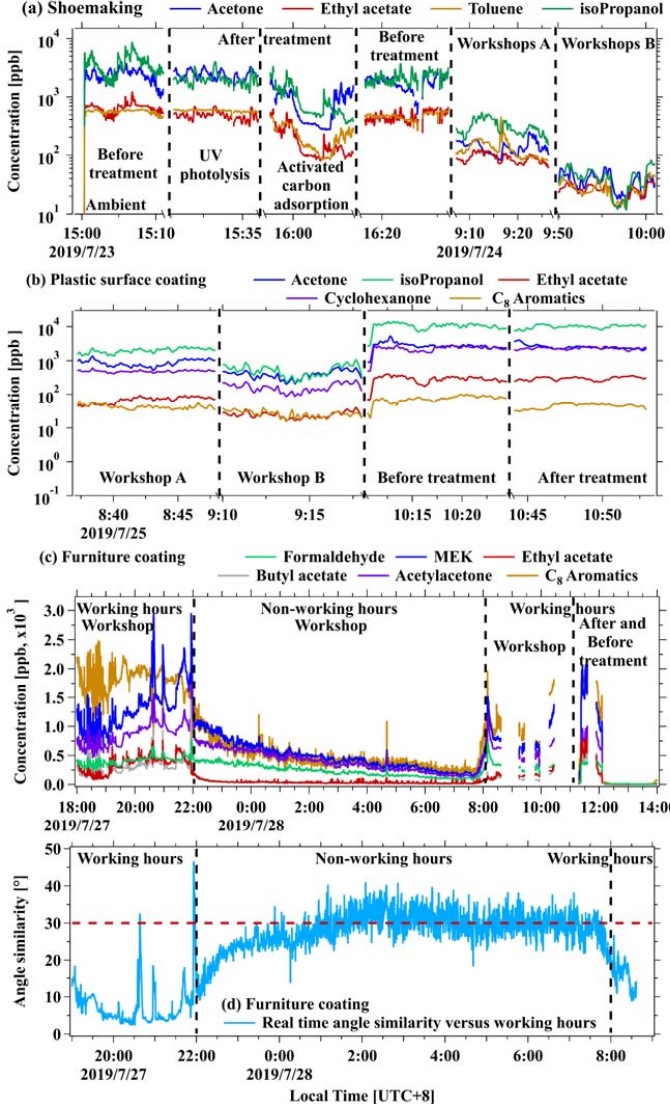

**Figure 1.** Real-time concentrations of representative ROGs from workshops, before and after the ROG treatment devices in (a) shoe making industry and (b) plastic surface coating industry, and (c) during working hours or non-working hours in the furniture coating industry. (d) The $\theta$ angles of mass spectra among real-time concentrations versus average concentration during working time (19:00-22:00) in the furniture coating industry.

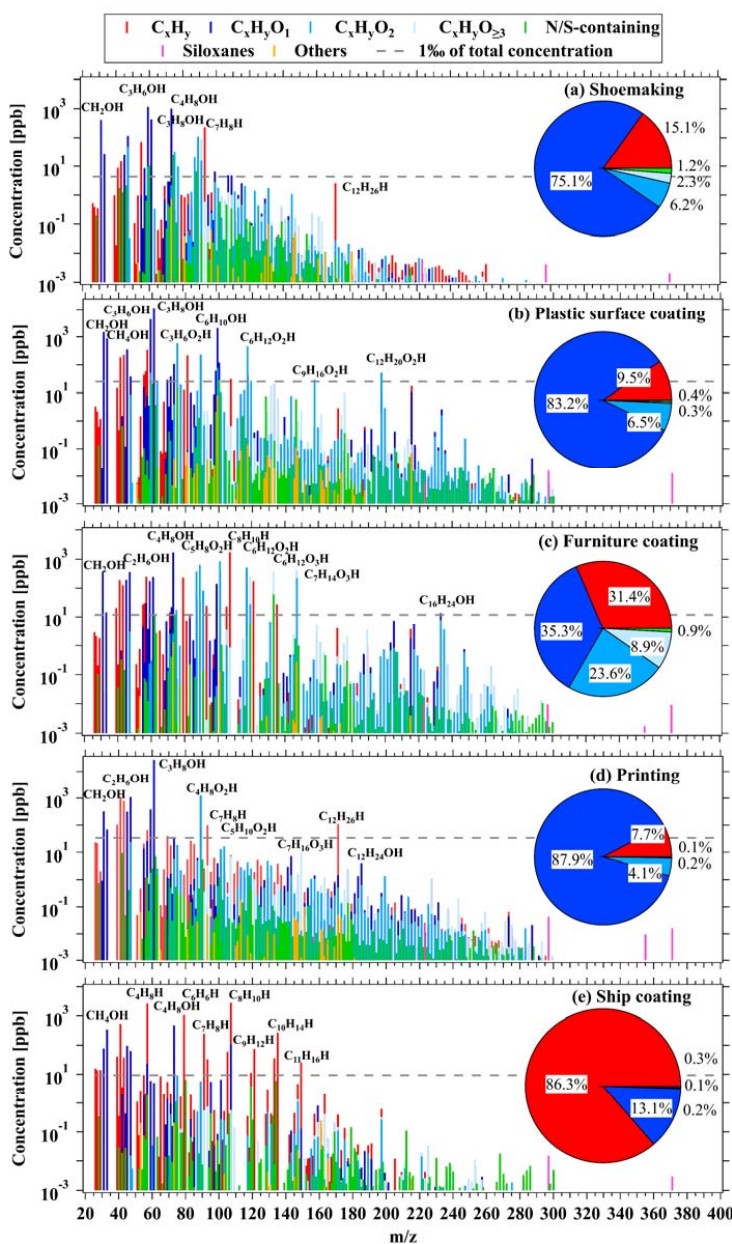

**Figure 2.** Average concentrations and fractions of ROGs measured by PTR-ToF-MS from stack emissions of (a) shoemaking, (b) plastic surface coating, (c) furniture coating, (d) printing, and (e) ship coating industries.



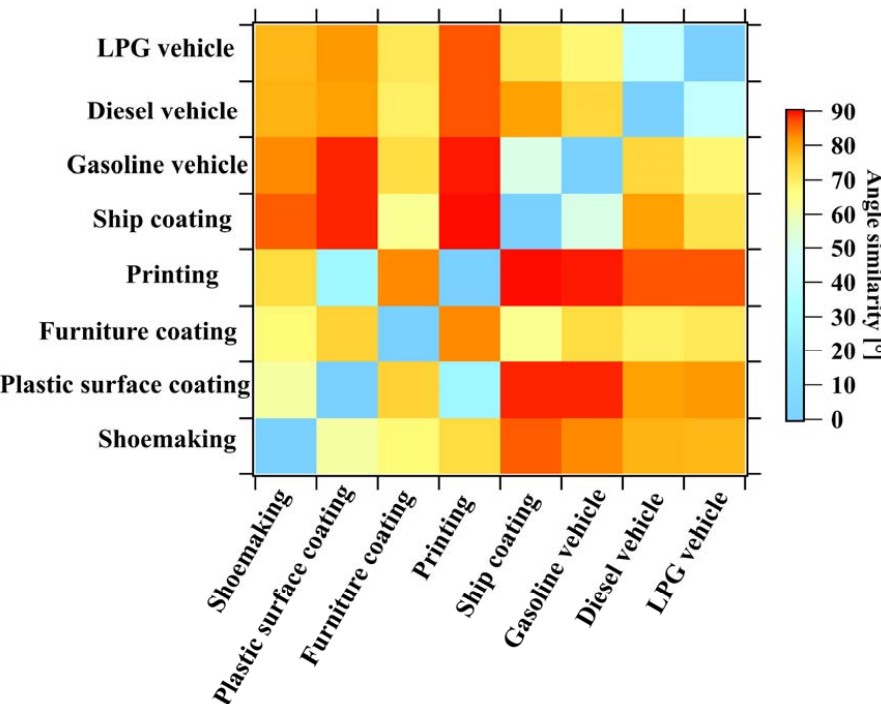


**Figure 3.** The $\theta$ angles (°) among the mass spectra of industrial VCP sources in this study and vehicular emissions from previous study (Wang et al., 2022).






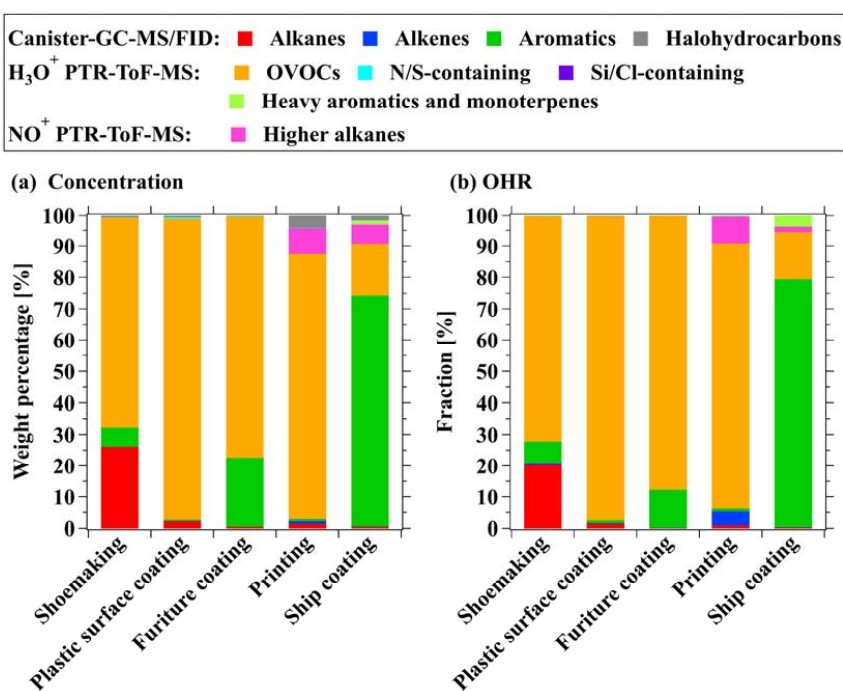

**Figure 4.** Fractions of (a) concentrations and (b) OHR for ROG components to total ROGs from stack emissions from shoemaking, plastic surface coating, furniture coating, printing, and ship coating industries.



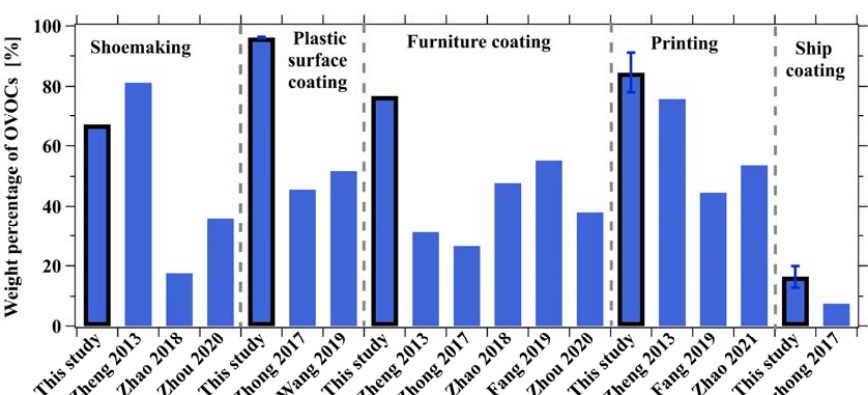

1053

**Figure 5.** Comparison of OVOC fractions determined from stack emission of industrial VCP sources in this study and those in previous studies. Error bars represent the standard deviations of the weight percentage of OVOCs.

1057



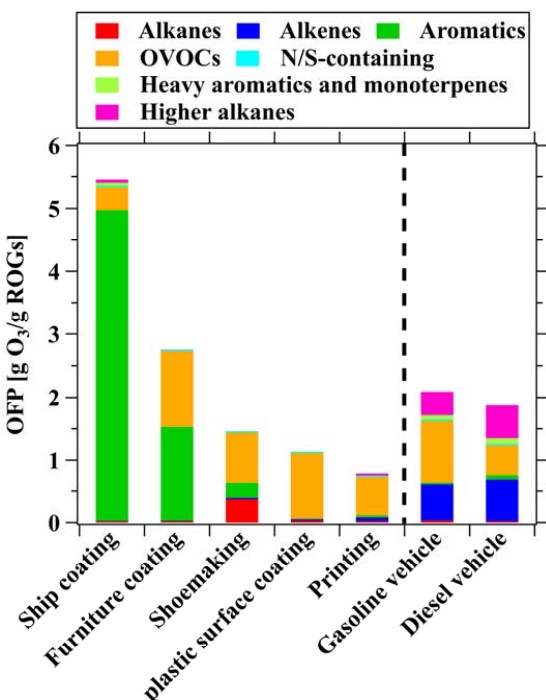

**Figure 6.** Comparison of OFP among various industrial VCP sources in this study and vehicular emissions from previous study (Wang et al., 2022).

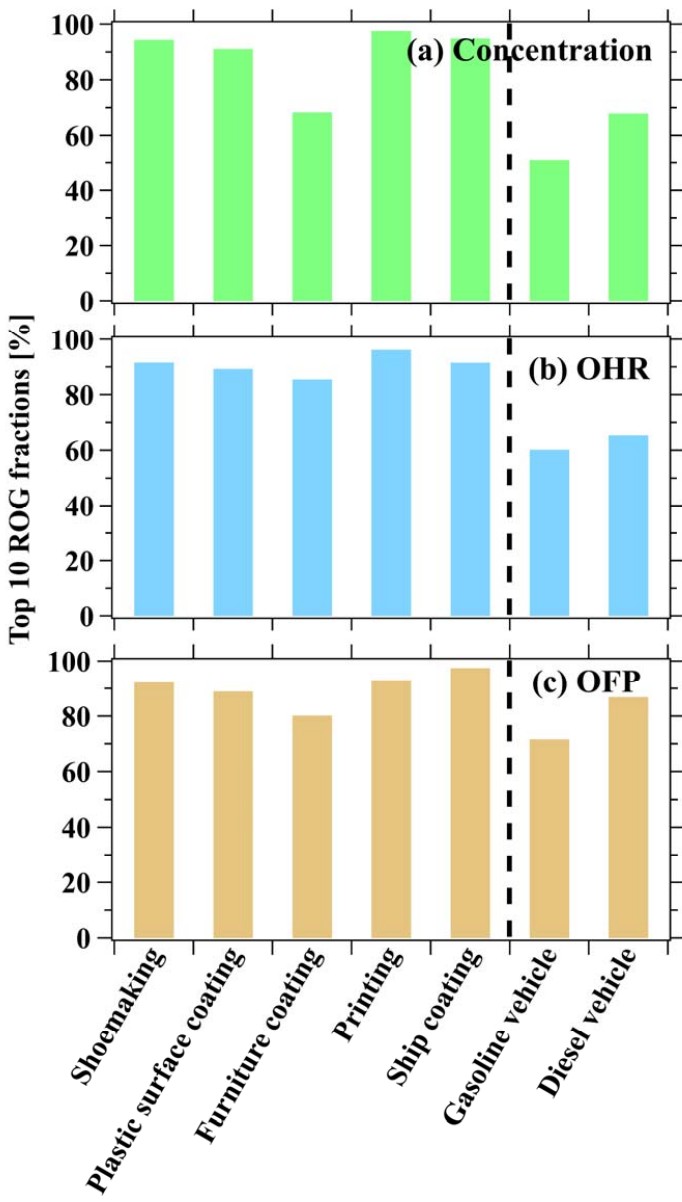


**Figure 7.** Accumulated fractions of the top ten species in total (a) ROG emissions, (b) OHR, and (c) OFP from industrial VCP sources in this study and vehicular emissions from previous study (Wang et al., 2022).









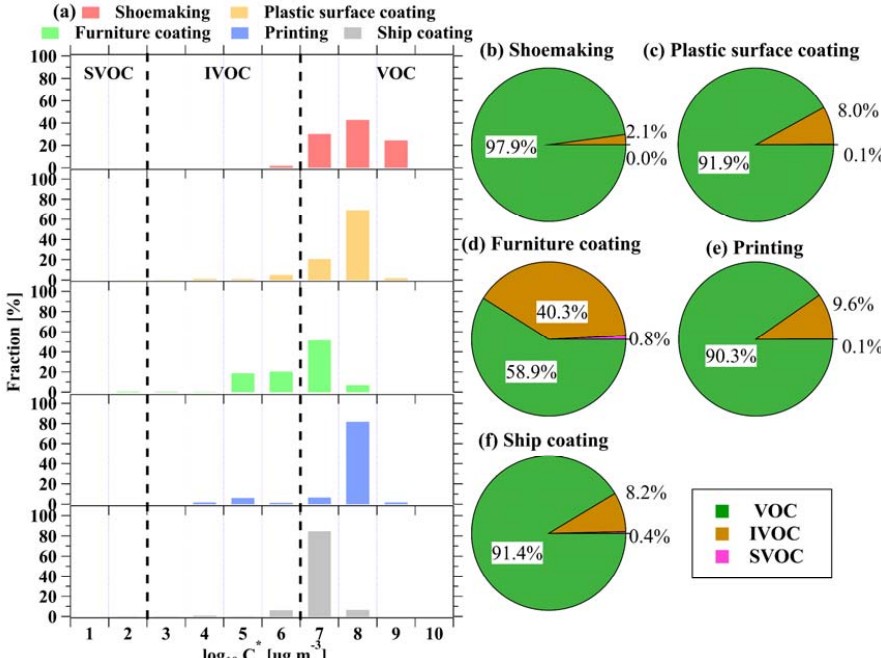


**Figure 8.** (a) Volatility-binned fractions from stack emissions of various industrial VCP

sources, and volatility-binned fractions in different ROG categories from stack

emissions of (b) shoemaking, (c) plastic surface coating, (d) furniture coating, (e)

printing, and (f) ship coating industries.

1072

1073

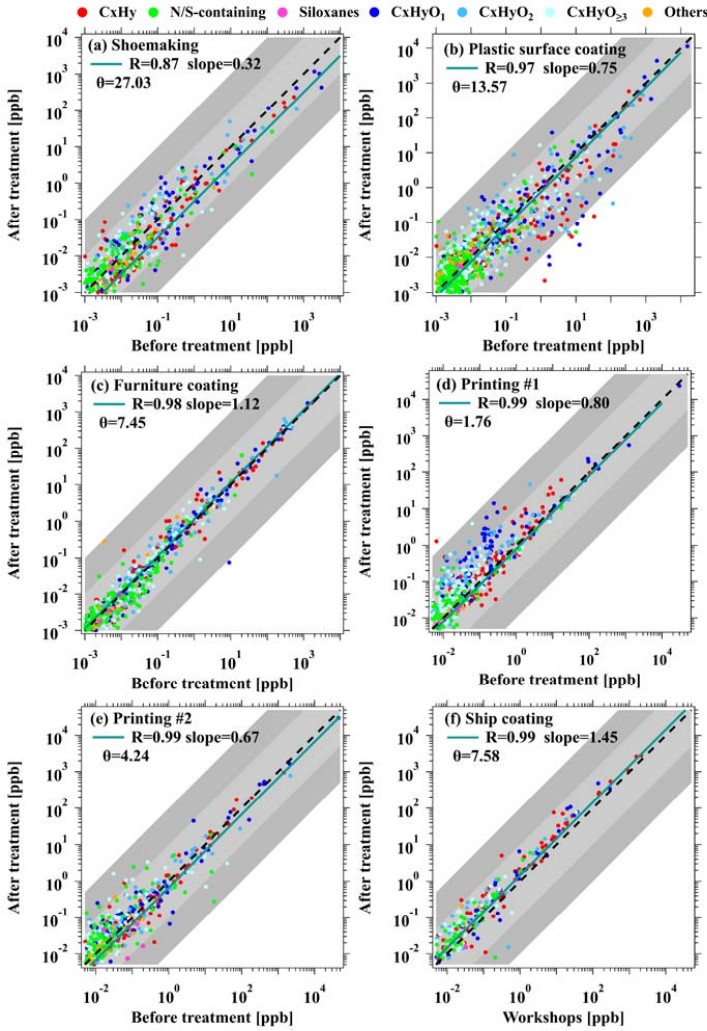

**Figure 9.** Scatterplots of ROG concentrations between before and after treatment with activated carbon adsorption + UV photolysis for (a) shoemaking, (b) plastics surface coating, (c) furniture coating, and (d) printing industries. Scatterplots of ROG concentrations between before and after treatment with catalytic combustion for (e) the printing industry, and ROG concentrations between workshops and after treatment with catalytic combustion (f) the ship coating industry. The green lines are the fitted results for all data points. The black dashed lines represent 1:1 ratio, and the shaded areas represent ratios of a factor of 10 and 100.

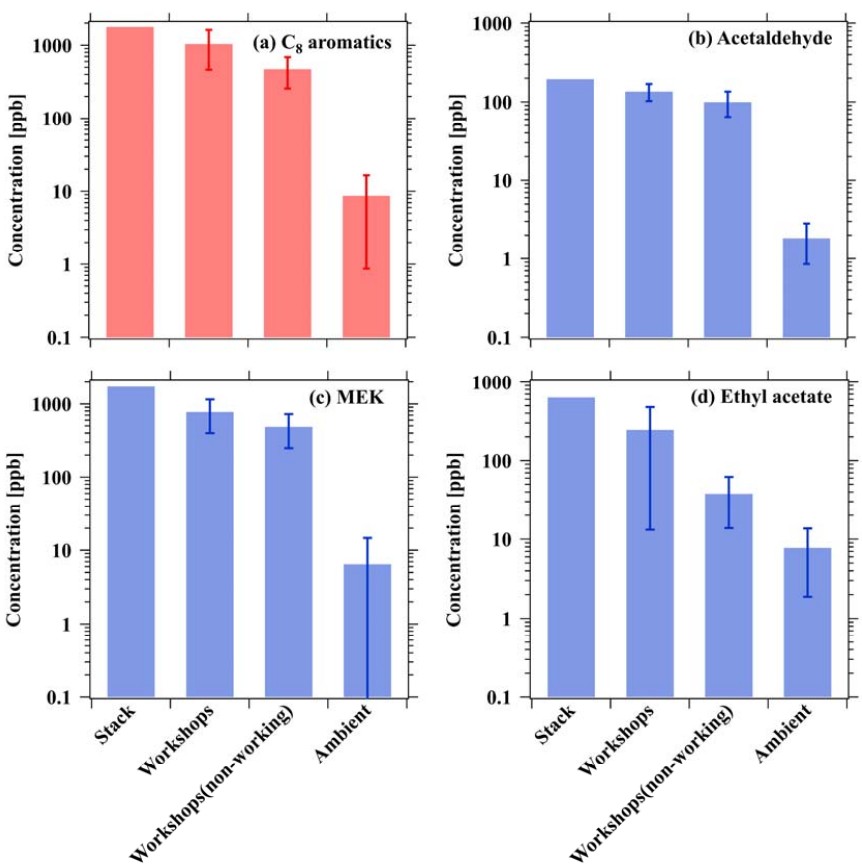

1084

**Figure 10.** Average concentrations of (a) C$_8$ aromatics, (b) acetaldehyde, (c) MEK, and
(d) ethyl acetate emission from the stack, workshops during working and non-working
hours in the furniture coating industry and ambient measurement near the industry,
respectively. Error bars represent the standard deviations of the concentration.

1089



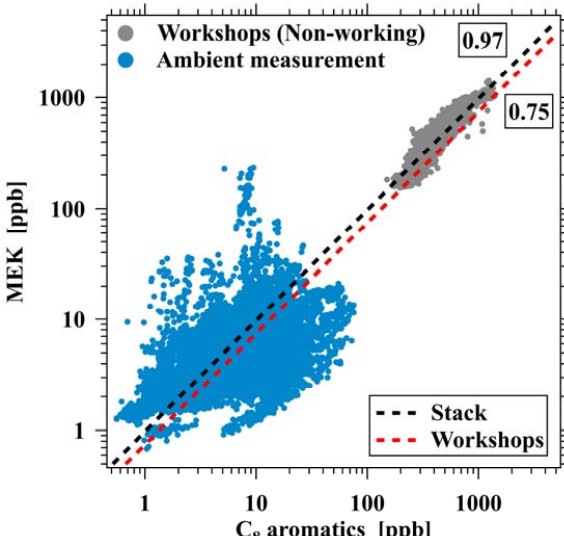

**Figure 11.** Scatterplot of MEK versus $C_8$ aromatics at workshops during non-working hours in the furniture coating industry and ambient measurement near the industry. The black and red dashed line represent ratios of ROG pairs for stack and workshops emission in furniture coating industry.