# Peer review of "Emission characteristics of reactive organic gases from"

_EGUsphere, 2024_

## Author Comment (AC1)

**Response to Reviewer #1's comments**

*The authors present comprehensive speciation of ROG emissions from industrial VCP sources, including shoemaking, plastic surface coating, furniture coating, printing, and ship coating industries. They use the combination of a PTR-ToF-MS in $H_3O^+$ and $NO^+$ to capture OVOCs and long chain alkanes, respectively, alongside a GC-MS to identify individual molecules and smaller alkanes. They highlight the important contribution of OVOCs not only to the overall emissions but also to the reactivity, ozone, and SOA formation potential. Finally, they evaluate the performance of ROG treatment devices used to reduce ineffectively emissions from these VCP sources. This paper provides unique insights on the emission fingerprint of VOCs and OVOCs from industrial VCPs in China and is suitable for publication after the following minor comments.*

Reply: We would like to thank you for your insightful comments, which helped us tremendously in improving the quality of our work. Please find our responses to individual comments below.

*Comments:*

*1.I recommend that the authors thoroughly proofread and improve the English, especially in the supplement and captions.*

Reply: Thanks for your suggestion. We have checked all these comments and checked the grammar and syntax throughout the manuscript and the supplement.

*2.Line 38-39: The meaning of this sentence is unclear to me.*

Reply: Thanks for your suggestion. We have re-wording this sentence.

The sentence in the Abstract (line 39-41) is modified to:

**We find that a few species can contribute the majority of the ROG emissions, and also their OHR and OFP from various industrial VCP sources.**

*3.Line 66: Delete the word "on."*

Reply: We have deleted "on".

30

*4.Lines 97-98: This sentence feels somewhat out of place.*

Reply: Thanks for your suggestion. We have re-wording this sentence.

The sentence in the Introduction (line 103-104) is modified to:

**More evidence shows that the contribution of VCP sources to anthropogenic ROG emissions is gradually becoming more prominent**.

36

*5.Line 100: Add "...attributed to a VCP-dominated..."*

Reply: We have added "a" in this sentence.

39

*6.Line 126: I'm unsure about the definition of ROG treatment devices. It would be helpful to define this term early on, as it is used extensively throughout, including in figure captions.*

Reply: Thanks for your suggestion. We agree with you that the definition of ROG treatment devices should be much early in our manuscript. We have removed descriptions in the Section 3.1 and Section 3.3, and added some description in the Section 2.1. We have modified them accordingly.

The sentence in the Introduction (line 131-135) is modified to:

**We investigated emission characteristics of ROGs across these industries, and utilized the dataset to analyze the contributions of different ROG components to total ROG emissions, OH reactivity (OHR), ozone formation potential (OFP), and volatility in various industrial VCP sources.**

The sentences in the Section 2.1 (line 150-162) are modified to:

**Typically, workshop waste gases are routed through collection devices (e.g. gas-collecting hoods, airtight partitions), and then processed in ROG treatment devices (e.g. ultraviolet-ray (UV) oxidation, activated carbon adsorption, combustion, and biodegradation). These treated gases are then released into the atmosphere through exhaust stacks. ROG treatment devices play a crucial role in reducing ROG emissions by employing recovery and destruction technologies**

**(Wang et al., 2023;Kamal et al., 2016). Recovery processes involve enriching and separating VOCs by means of temperature or pressure changes and selective absorbents, while destruction processes converts VOCs into harmless substances such as $CO_2$ and $H_2O$ through combustion (Wang et al., 2023). In this study, we evaluate two types of ROG treatment devices: activated carbon adsorption combined with UV photolysis devices (installed in shoemaking, plastic surface coating, furniture coating, and printing industries) and catalytic combustion devices (installed in printing and ship coating industries).**

*7.Line 149: These sampling lines are quite long. Is there any treatment for wall losses, especially for sticky OVOCs?*

Reply: Thanks for your suggestion. The sampling tubings used varied in length from 10 m to 50 m for most sampling sites. A tubing with a length of 100 m were employed for sampling at the ROG treatment device at the shoemaking industry, as the treatment device is located on the 9th floor of the building. During the campain, all sampling tubing were shielded with aluminum foil during the campaign. To investigate the potential wall losses resulting from the use of long tubes, we conducted an assessment of the uncertainty related to the sampling techniques in laboratory tests. The results indicated that the tubing had a minimal impact on most ROG species, affirming the feasibility of measurements using the long PFA tubing. Further details can be found elsewhere (Li et al., 2023).

The sentences in the Section 2.1 (line 169-172) are modified to:

**The use of long tubing was assessed through laboratory tests, which showed that the tubing had a negligible and minor influence on most ROG species. This confirmed the feasibility of measurement using long PFA tubing, more detail can be found elsewhere (Li et al., 2023).**

*8.Line 188: Remove the double dot.*

Reply: We have removed the double dot.

88

9.*Lines 221-223: This sentence is unclear. Please rephrase it.*

Reply: Thanks for your suggestion. Considering that this sentence seem to be abrupt here, we removed it in the revised manuscript.

92

10.*Lines 235-238: One could argue that this agreement is not ideal when both axes are in logarithmic scale. The differences are often greater than a factor of 2. More discussions on these differences would be great, especially considering the fragmentation interferences for both ionization methods.*

Reply: Thanks for your suggestion. The correlation between two modes are slightly weaker than the ambient measurements reported in our previous study. Here, we have added some descriptions about the differences between both ionization methods.

The sentences in the Section 2.3 (line 255-260) are modified to:

**Finally, the comparison between PTR-ToF-MS with $H_3O^+$ and $NO^+$ chemical ionization is shown in Fig. S4-S5. Previous studies have shown good agreement between measurements obtained using PTR-ToF-MS with $H_3O^+$ and $NO^+$ chemical ionization in ambient measurements (Wang et al., 2020). However, a slightly weaker correlation was observed in industrial VCP sources, potentially due to the large changes for different species between the switch of the two reagent ions.**

109

11.*Section 3.1: The current organization of the paper, with frequent references to Figures 1 and 2, makes it difficult for the reader to follow. I suggest reconstructing the paper to address each emission source separately, with overview graphs that combine information from both (or more) figures. The comparison of all sources could be presented in a separate figure.*

Reply: Thanks for your suggestion. We have reconstructed Section 3.1 by added subheadings (3.1.1 and 3.1.2), each emission source is now individually introduced in

117 Section 3.1.1. In addition, we presented combine information from Fig. 1 and Fig. 2

118 could better compare the emission characteristics of various emission sources, and we

119 have included a graph (Fig. S9) to compare the fraction of different ion categories

120 measured by the PTR-ToF-MS across various industrial VCP sources.

121   The subheadings in the Section 3.1(line 280-382) is modified to:

122 **3.1.1 Emission characteristics from industrial VCP sources**

123 **A. Shoemaking industry**

124 **B. Plastic surface coating industry**

125 **C. Furniture coating industry**

126 **D. Printing industry**

127 **E. Ship coating industry**

128 **3.1.2 Comparison of ROG composition from industrial VCP sources**

129   The sentence in the Section 3.1 (line383-384) is modified to:

130 **The quantification of the proportions of different ion categories measured**

131 **by the PTR-ToF-MS across various industrial VCP sources is shown in Fig.2 and**

132 **Fig. S9.**

[Figure]

133

134 **Figure S9. The fractions of different ROG categories measured by the PTR-ToF-**

135 **MS from stack emissions across various industrial VCP sources.**

12.*Lines 251-252: Please provide more elaboration on what is meant here. Is it that the angle θ approach was previously only used for AMS spectra, which have substantial fragmentation compared to PTR?*

Reply: Thanks for your suggestion. We changed this part:

The sentences in the Section 2.4 (line275-278) are modified to:

**As these previous studies utilize the similarity analysis on mass spectra of aerosol mass spectrometer (AMS) obtained from electron ionization, leading to very similar mass spectra for different sources.**

13.*Lines 260-262: It's unclear what is meant by collection devices and collection and treatment devices, as well as their differences.*

Reply: Thanks for your suggestion. We have added some descriptions about these two devices in the Section 2.1 in the revised manuscript.

The sentences in the Section 2.1 (line 150-154) are modified to:

**Typically, workshop waste gases are routed through collection devices (e.g. gas-collecting hoods, airtight partitions), and then processed in ROG treatment devices (e.g. ultraviolet-ray (UV) oxidation, activated carbon adsorption, combustion, and biodegradation). These treated gases are then released into the atmosphere through exhaust stacks.**

14.*Lines 263-264: How do these treatment devices work? How do they ensure fewer ROG emissions?*

Reply: Thanks for your suggestion. We have added some descriptions about the operation of treatment devices. in the Section 2.1 in the revised manuscript.

The sentences in the Section 2.1 (line154-159) are modified to:

**ROG treatment devices play a crucial role in reducing ROG emissions by employing recovery and destruction technologies (Wang et al., 2023;Kamal et al., 2016). Recovery processes involve enriching and separating VOCs by means of**

temperature or pressure changes and selective absorbents, while destruction processes converts VOCs into harmless substances such as $CO_2$ and $H_2O$ through combustion (Wang et al., 2023).

15.Lines 265-266: I'm unsure about the meaning of this sentence. Please define what stacked emissions are.

Reply: Thanks for your suggestion. We have modified this description here.

The sentence in the Section 3.1 (line 285-287) is modified to:

As the waste gas was directly discharged into the ambient air from exhaust stacks, the after treatment emission can be considered as stack emission (Zheng et al., 2013).

16.Line 327: Replace "are" with "have".

Reply: We have replaced "are" with "have".

17.Lines 327-331: Did the authors measure outside air and consider its influence on the measured spectra? Could there be any influence from outside air on the factory spectra? Given that later on, ambient measurements are discussed, it might be worth comparing the two in more detail.

Reply: Thanks for your suggestion. We have measured outside air before and after each industrial VCP source. We compared the similarity between the mass spectra obtained during non-working hour and those for outside air. As shown in Fig.S8, the results indicate the outside air has almost no influence on ROG emissions during non-working hours. We have added some descriptions in the Section 3.1.

The sentence in the Section 3.1 (line 351-354) is modified to:

Additionally, the poor similarity observed between real-time concentrations in workshops during non-working hours and those in the outside air suggests that outside air has minimal influence on ROG emissions during non-working hours (Fig. S8).

[Figure]

**Figure S8. The *θ* angles of mass spectra among real-time concentrations versus outside air measurement in the furniture coating industry.**

*18.Line 354: Change to "a quantification."*

    Reply: We have replaced "quantified" with "quantification ".

*19.Line 427: Delete the word "in."*

    Reply: We have deleted "in".

*20.Lines 453-457: It would be helpful to provide a more detailed description of how O₃ sensitivity was calculated here, rather than limiting it to previous citations.*

    Reply: Thanks for your suggestion. We have added some descriptions in the Section 3.2.

    The sentences in the Section 3.2 (line 481-488) are modified to:

**To facilitate for making controlling strategies of ozone, we determine the OFP from a unity of emission from different sources for comparison (Yuan et al., 2010;Na and Pyo Kim, 2007), which represent the ability to ozone formation from ROG sources on a relative basis (Fig. 6), and calculated using the following equation:**

$$OFP_i = \sum_{j=1}^{n} f_{ji} MIR_j \qquad (2)$$

Where $OFP_i$ is the estimated ozone formation amount when 1 g ROG is emitted from source i, $f_{ji}$ is the mass fraction of species j in source i, and $MIR_j$ is the maximum incremental reactivity (MIR) of species j (Carter, 2007).

*21.Lines 536-530: The definition of treatment devices should be introduced earlier and discussed at the beginning of the paper.*

Reply: Thanks for your suggestion. We agree with you that the definition of ROG treatment devices should be much early in our manuscript. We have removed descriptions in the Section 3.1 and Section 3.3, and added some description in the Section 2.1. We have modified them accordingly.

The sentences in the Section 2.1 (line150-162) are modified to:

**Typically, workshop waste gases are routed through collection devices (e.g. gas-collecting hoods, airtight partitions), and then processed in ROG treatment devices (e.g. ultraviolet-ray (UV) oxidation, activated carbon adsorption, combustion, and biodegradation). These treated gases are then released into the atmosphere through exhaust stacks. ROG treatment devices play a crucial role in reducing ROG emissions by employing recovery and destruction technologies (Wang et al., 2023;Kamal et al., 2016). Recovery processes involve enriching and separating VOCs by means of temperature or pressure changes and selective absorbents, while destruction processes converts VOCs into harmless substances such as $CO_2$ and $H_2O$ through combustion (Wang et al., 2023). In this study, we evaluate two types of ROG treatment devices: activated carbon adsorption combined with UV photolysis devices (installed in shoemaking, plastic surface coating, furniture coating, and printing industries) and catalytic combustion devices (installed in printing and ship coating industries).**

*22.Section 3.3: This discussion is based on a double logarithmic graph that shows a highly variable scatter by a factor of 10 to 100. In many cases, most compounds are increasing, not just the ones highlighted by the authors. It would be beneficial for the*

*authors to provide a more detailed analysis for this section. They should describe the trends by group of compounds and dive into the reasons for the observed differences, supported by clear graphs indicating the efficiency of the treatment devices e.g., histogram percentage differences per source category.*

Reply: Thanks for your suggestion. We added a graph depicting treatment efficiencies of various groups of ROGs from industrial VCP sources as Fig. S11. These treatment efficiencies are obtained from the slope of each group (Fig. R1). The analysis reveals an overall increase in most groups of ROGs, we have revised the descriptions in Section 3.3 accordingly.

The sentence in the Section 3.3 (line 567-569) is modified to:

**Nonetheless, it is evident that the treatment efficiency has not reached the desired levels for all ROG groups (Fig. S11), which possibly due to the challenges associated with effectively removing majority ROG emissions using current treatment technologies.**

The sentences in the Section 3.3 (line 578-583) are modified to:

**The lowest treatment efficiency of ROG was obtained in the furniture coating industry (slope=1.12). This treatment device demonstrates inefficiency for all ROG groups (Fig. S11). The inadequate performance of the ROG treatment devices in this specific facility may be attributed to a number of possible reasons, e.g., delayed replacement of activated carbon and other adsorption materials, and the implementation of the UV photolysis device could potentially result in the generation of more ROGs as byproducts.**

[Figure]

266

**Figure S11. Treatment efficiencies of different ROG categories provided by treatment devices in various industrial VCP sources.**

[Figure]

269

Figure R1. Scatterplots of (a) $C_xH_y$ ions and (b) $C_xH_yO_{\geq 3}$ ions concentrations between before and after treatment for the plastic surface coating industry. The brown and blue lines are the fitted results for $C_xH_y$ ions and $C_xH_yO_{\geq 3}$ ions data points. The black dashed lines represent 1:1 ratio, and the shaded areas represent ratios of a factor of 10 and 100.

274

*23. Line 591: Delete the word "in."*

Reply: We have deleted "in".

*24. Lines 612-614: It would be valuable if the authors could verify these ratios by running a PMF on the ambient data. Observing whether they can separate different sources and extract the aromatic to MEK ratio would provide more confidence in using this ratio as an indicator of different VCP emissions. Was the site downwind of the industry? Meteorological data could also help narrow the influence of the different industrial sectors.*

Reply: Thanks for your suggestion. In this section, we observed that the concentration of selected ROG species is significantly higher than those reported in previous studies conducted in other environments. The peak concentrations of MEK exceeding 200 ppb from the ambient measurements are among the highest reported in the literature. The MEK / $C_8$ aromatics ratio can serve as good evidence for the impact of industrial VCP sources on ambient measurements in industrial areas. Due to a lack of meteorological data, we are unsure whether the site is downwind of the industry in this study. However, the consistency in concentrations of MEK and $C_8$ aromatics suggests a substantial influence of industrial VCP sources on ROG emissions in industrial areas. We have modified Fig. 10 and some disscuss on Fig. 11, and added Fig. S13 to compare the concentration of MEK and $C_8$ aromatics bwtween our study and previous studies. We have modified these comments accordingly.

The title of the Section 3.4 (line 603) is modified to:

**3.4 Impact of industrial VCP sources on ambient air**

The sentences in the Section 3.4 (line 641-649) are modified to:

[revised manuscript text omitted]

---

## Author Comment (AC2)

**Response to Reviewer #2's comments**

*This work investigated the emissions of ROGs from five industrial VCP sources in China, including shoemaking, plastic surface coating, furniture coating and shipping coating industries. PTR-ToF-MS and GC-MS/FID are combined together to develop comprehensive speciation of VOC from these industrial sources in PRD, China. The manuscript is generally well organized. Some statements are unclear and need to be clarified. I also suggest authors polish English and grammar throughout the manuscript. Please see below for my detailed comments.*

Reply: We would like to thank you for your insightful comments, which helped us tremendously in improving the quality of our work. We have checked the grammar and syntax throughout the manuscript and the supplement. Please find our responses to individual comments below.

*1.Abstract: This work is only for PRD, China, instead of the whole nation. Please clarify this in the title and abstract to avoid misunderstanding.*

Reply: Thanks for your suggestion. We have modified this description in the title and abstract.

The title in the revised manuscript is modified to:

**Emission characteristics of reactive organic gases from industrial volatile chemical products (VCPs) in the Pearl River Delta (PRD), China**

The sentence in the Abstract (line 21-24) is modified to:

**This study aimed to investigate the emissions of ROGs from five industrial VCP sources in the Pearl River Delta (PRD) region of China, including shoemaking, plastic surface coating, furniture coating, printing, and ship coating industries.**

*2.Line 30: Not sure what this sentence means. Please keep in mind that this study doesn't cover all emission sources. Please clarify this sentence to avoid misunderstanding.*

Reply: Thanks for your suggestion. We have modified this description here.

The sentence in the Abstract (line 30-32) is modified to:

**Moreover, mass spectra similarity analysis revealed significant dissimilarities among the ROG emission from industrial activities, indicating substantial variations between different industrial VCP sources.**

*3.Line 32: so, what's the proportion of OVOCs for ship coating industry then? Does it make big difference using solvent-borne coatings or waterborne coatings for OVOC proportion?*

Reply: Thanks for your suggestion. There is a significant difference in the proportion of OVOCs between solvent-borne coatings and water-borne coatings. We have included additional descriptions regarding the proportion of OVOCs in the ship coating industry.

The sentence in the Abstract (line 32-36) is modified to:

**Except for the ship coating industry utilizing solvent-borne coatings, the proportions of OVOCs range from 67% to 96% in total ROG emissions and 72% to 97% in total OH reactivity (OHR) for different industrial sources, while the corresponding contributions of OVOCs in the ship coating industry are only 16%±3.5% and 15%±3.6%.**

*4.Line 37-39: please improve the statement.*

Reply: Thanks for your suggestion. We have re-wording this sentence.

The sentence in the Abstract (line 39-41) is modified to:

**We find that a few species can contribute the majority of the ROG emissions, and also their OHR and OFP from various industrial VCP sources.**

*5.Line 41: Why is the treatment efficiency negative?*

Reply: Thanks for your suggestion. The negative treatment efficiency of ROG was obtained in the furniture coating industry, as shown in the discussion in Section 3.3. This treatment device demonstrates inefficiency for all ROG groups. The inadequate performance of the ROG treatment devices in this specific facility may be attributed to a number of possible reasons, e.g., delayed replacement of activated carbon and other adsorption materials, and the implementation of the UV photolysis device could potentially result in the generation of more ROGs as byproducts.

*6.Line 74: Not accurate statement. The substitution of solvent-borne VCPs by water-borne ones are for several sources., e.g., interior wall painting.*

Reply: Thanks for your suggestion.  The substitution of solvent-borne VCPs by water-borne VCPs are not for all of industrial VCP sources. We have revised descriptions here.

The sentences in the Introduction (line 74-81) are modified to:

**To mitigate the emissions of most primary pollutants, stricter emission standards have been implemented along with advancements in ROG treatment technologies in China. Specifically, water-borne VCPs has substituted solvent-borne VCPs in several industries, such as printing, interior wall coating, and automotive manufacturing. However, the replacement in steel structures, automotive plastic parts manufacturing and ship building industries remains below 3% (Mo et al., 2021;Li et al., 2019;Shi et al., 2023;Wang et al., 2023).**

*7.Line 159: I'm curious how to combine PTR-ToF-MS with GC-MS/FID measurements when they overlap? How to handle the un-known species?*

Reply: Thanks for your suggestion. Careful consideration should be given to the overlap of ROG species in the combined measurements of PTR-ToF-MS and GC-MS/FID, to make sure that each species should only be considered once. Species that are not calibrated were semi-quantified using methods based on the kinetics of proton-transfer reactions of $H_3O^+$ with ROGs (Fig. S2). We have added some descrition about these in Section S2 in the Supplement.

The sentence in the Setion 2.1 (line 180-181) is modified to:

**The selection of overlapping ROGs was similar to a previous study (Table. S2).**

The sentences in the Setion S2 in the Supplment (line 85-92) are modified to:

**In this study, a more comprehensive speciation of ROGs was achieved by**

**the combination of GC-MS/FID and PTR-ToF-MS, the same ROG species from**

**the combination measurement should be counted only once. All ROG species**

**detected in this study is summarized in Table S2. Specifically, to facilitate**

**comparison with traditional photochemical assessment monitoring stations**

**(PAMS) species, $C_6$-$C_{10}$ aromatics were identified using GC/MS-FID, while $C_{10}$-**

**$C_{12}$ alkanes were detected using $NO^+$ PTR-ToF-MS, as GC-MS/FID only**

**containing the n-alkanes. For unknown ROG species, we used the semi-quantity**

**based on the methods.**

**Table S2. Detailed information of ROG species measured by different instruments.**

| Components | Measurements | ROG species |
|---|---|---|
| OVOCs | PTR-$H_3O^+$ | formula only including CHO |
| N/S-containing | PTR-$H_3O^+$ | formula including CHN, CHS, CHON, CHOS, and CHONS |
| Heavy aromatics and monoterpenes | PTR-$H_3O^+$ | monoterpenes, $C_{11}$-$C_{20}$ aromatics, and polynuclear aromatic hydrocarbons (PAHs) |
| Higher alkanes | PTR-$NO^+$ | $C_{10}$-$C_{20}$ acyclic, cyclic and bicyclic alkanes |
| Alkanes | GC-MS/FID | $C_2$-$C_9$ alkanes |
| Alkenes | GC-MS/FID | $C_2$-$C_6$ alkenes |
| Aromatics | GC-MS/FID | $C_6$-$C_{10}$ aromatics |
| Halohydrocarbons | GC-MS/FID | $C_1$-$C_6$ halohydrocarbons |

*8.Line 299: have you found any additional important OVOCs using PTR-ToF-MS?*

*Please list them or at list some examples here.*

Reply: Thanks for your suggestion. In the Introduction, we have discussed the utilization of PTR-ToF-MS to enhance the characterization of OVOC emissions from industrial VCPs. In addition, some OVOCs with high concentrations (i.e. acetates and acrylates) have been listed in the line 317-320 in the revised manuscript (line 294-296

in the original manuscript), which were seldom reported in previous studies.

  Considering that this sentence seem to be abrupt here, we removed it in the revised manuscript.

**Reference:**

[revised manuscript text omitted]